# Benchmarking the Effect of Poisoning Defenses on the Security and Bias of the Final Model

**Nathalie Baracaldo**[1]**, Kevin Eykholt**[1]**, Farhan Ahmed**[1]**, Yi Zhou**[1]**, Shriti Priya**[1]**, Taesung Lee**[1]**, Swanand Kadhe**[1]**, Mike Tan**[2]**, Sridevi Polavaram**[2]**, Sterling Suggs**[3]**, Yuyang Gao**[4]**, David Slater**[3]

[1]IBM Research
baracald@us.ibm.com, {kheykholt, farhan.ahmed, yi.zhou, shritip, taesung.lee, swanand.kadhe}@ibm.com
[2]The MITRE Corporation
{ytan, spolavaram}@mitre.org
[3]Two Six Technologies
{sterling.suggs, david.slater}@twosixtech.com
[4]Emory University
yuyang.gao@emory.edu

## Abstract

Machine learning models are susceptible to a class of attacks known as adversarial poisoning where an adversary can maliciously manipulate training data to hinder model performance or, more concerningly, insert backdoors to exploit at inference time. Many methods have been proposed to defend against adversarial poisoning by either identifying the poisoned samples to facilitate removal or developing poison agnostic training algorithms. Although effective, these proposed approaches can have unintended consequences on other aspects of model performance, such as worsening performance on certain data sub-populations, thus inducing a classification bias. In this work, we evaluate several adversarial poisoning defenses. In addition to traditional security metrics, i.e., robustness to poisoned samples, we propose a new metric to measure the potential undesirable discrimination of sub-populations resulting from using these defenses. Our investigation highlights that many of the evaluated defenses trade decision fairness to achieve higher adversarial poisoning robustness. Given these results, we recommend our proposed metric to be part of standard evaluations of machine learning defenses.

## 1 Introduction

Machine learning (ML) is used in numerous critical applications including healthcare, finance, and the Internet-of-Things. However, the sensitivity of these applications also motivates a need to develop secureML algorithms to avoid safety and security incidents. In particular, *adversarial poisoning attacks* on machine learning has received significant attention [18]. In a poisoning attack, an attacker modifies a portion of the training data to influence and/or degrade the performance of the trained model. Often, the goal is to encode a backdoor in a few training samples, which the attacker can later trigger at inference time. Despite the presence of poisoned training samples, the overall performance on benign (i.e., non-trigger) inputs is often satisfactory, thus avoiding suspicion the model has been poisoned. However, when a sample appears containing a backdoor trigger (e.g., image patch), the desired erroneous behavior occurs (e.g., a targeted misclassification of a malicious input).

2022 Trustworthy and Socially Responsible Machine Learning (TSRML 2022) co-located with NeurIPS 2022.

To mitigate the effect of adversarial poisoning, a variety of defense techniques have been proposed: flagging the data that is "suspicious" [3, 21], segmenting training data using model ensembles [15, 24], abstaining from predicting [23], or by manipulating the training data before feeding it to the training process to neutralize potential malicious modifications [4, 9, 26, 27]. An ideal defense would fully mitigate or remove the effect of poisoning *without any undesirable side-effects*.

The side-effects of adversarial poisoning defenses have not been studied by prior works. Prior work primarily focused on reducing the poisoning attack success rate, while maintaining accuracy on benign samples. Although both of these metrics are useful in evaluating the quality of a prospective defense, they do not provide a complete picture of the final "defended" model's behavior. For example, we discovered most filtering defenses remove a consistent fraction of the training data regardless of whether the removed samples are poisonous or not. In general, as poisoned samples are a small subset of the training data containing a unique set of features, i.e. the trigger, we hypothesize that prior defenses appeared effective as they (possibly unintentionally) mitigated the influence of outlier features. However, outliers are frequently under-represented *benign* sub-populations that occur naturally in the data and can be harmed by outlier mitigation techniques, which may result in potential decision bias against benign sample. Typically, a sub-population within a class may include samples collected under specific conditions (e.g., a winter storm or lightening), a particularly under-represented sample (e.g., very uncommon type of airplane), or an under-represented minority (e.g., women). An example of benign sub-populations within a class is shown in Appendix A.1.

In this paper, we study the side-effects of existing poisoning defenses and show many defenses are non-ideal as they negatively affect the decision fairness of the defended model on under-represented benign sub-populations. We benchmark existing defenses against dirty and clean label poisoning attacks using traditional security metrics and a new metric that measures the side-effects of prospective defenses on the model's decision. Borrowing from fairness literature, we use the *Statistical Parity Difference (SPD)* [5], to measure the effect a defense has on *minority* sub-populations in the data (e.g., images collected with poor visibility in a driving dataset). Our extensive experimental results shed new light on the effects that existing poisoning defenses have on decision fairness.

## 2 Background: Poisoning Attacks and Defenses

### 2.1 Poisoning Attacks

Poisoning attacks manipulate a small percentage of the data used during training to achieve some adversarial goal, often backdoor injection. They can be broadly classified into *dirty label* and *clean label* attacks. In dirty label attacks [10, 16], the adversary's goal is to induce a misclassification into a target class through use of a backdoor trigger. The adversary generates a trigger (e.g. image patch) and poisons a percentage of the training data by adding the trigger, $X_p \subseteq X$, as well as modifying the labels of those samples, $Y_p \subset Y$, to the target class. At inference time, the adversary adds the trigger to induce the backdoor behavior, targeted misclassification. The attack is designed so that when the backdoor is not present, the model behaves normally.

In contrast to dirty label attacks, clean label attacks **do not modify** the labels and rely on more inconspicuous modifications. As poisoned inputs appear consistent with their labels, human inspection is unlikely to detect the attack. A large variety of clean label attacks have been proposed in literature [19]. Two distinct attacks are the original *Clean Label Backdoor Attack (CLBD)* [22] and *Witches' Brew* (also known as *Gradient Matching*) [7]. These attacks often solve optimization problems to generate poison images close to the original without changing the labels which inject the backdoor to the model if used for training. For example, a crafted cat image still looks like a cat to a human, but the model sees the image as if it is a truck in terms of features or training gradients. Turner et al. leverage a GAN and adversarial example approach to generate poison images while ensuring the perturbation is bounded. In Witches' Brew, the attacker chooses one or few images in the test data as *trigger images* and aims to make the model classify them as the target class(es). This attack applies bounded perturbations to the poison data by aligning the training gradient of the poison data with the correct labels, and that of the trigger samples with the target labels, using a surrogate model.

## 2.2 Poisoning Defenses

To mitigate poisoning attacks, multiple defenses have been proposed recently. We provide a high-level overview of the specific defenses that we evaluate in Section 4.

**Activation Defense** [3] is a filtering-based defense, which analyzes the training set and filters out samples that are deemed "too different" with respect to the rest of the data. After training, the training data is passed again through the model and the last layer activations are recorded and clustered. Samples associated with either *small* clusters or isolated clusters are removed from the training set. In our evaluation, potentially poisonous data was marked based on the smallest activation cluster(s).

**Spectral Signatures** [21] is a filtering-based defense where the activations of the network for each training sample are analyzed using singular value decomposition (SVD). Samples with unusual SVD are removed from the training set. This defense has an additional hyperparameter to define how much poison is expected to be in the training data. In our evaluation, we use the best case scenario where the expected poisoning hyperparameter exactly matches the true poisoning percentage.

**Deep Partition Aggregation (DPA)** [15] is an ensemble-based defense that creates multiple weak classifiers and performs inference by voting. The training samples are split in $k$ disjoint partitions $P_1, ..., P_k$ and each partition $P_i$ is used to train a different model to create an ensemble. During inference, the models are ensembled and a prediction is made based on a majority vote.

**Finite Aggregation** [24] is an extension of DPA and includes two hyper-parameters to guide the ensembling process, $k$ and $d$. They are defined as the inverse sensitivity and the spreading factor, respectively. First, the defense partitions the training dataset into $kd$ disjoint partitions. Then, each data partition is assigned to $d$ of the $kd$ submodels in the partition. The $kd$ submodels models are trained on their assigned partitions. During inference, a prediction is made based on a majority vote.

**Inverse Self-Paced Learning (ISPL)** [13] is a filtering defense that relies on identifying "compatible or homogeneous sets" in the training data. The defense defines the notion of "self-expanding sets" and propose an iterative approach, which results in groups of homogeneous sets, i.e. all of the samples in the set belong to either the primary or noisy distribution. In their scenario, the noisy distribution, which is assumed to be the minority, contains the poisoned data. Once the data has been segmented, they train a model on each partition and use each model to classify data from all of the other partitions. Using a majority voting scheme based on the misclassification rate on the other partitions, the primary and poison distributions can be identified.

**Adversarial Training** [17] was originally used as a defense against evasion attacks, but has been examined in some works as a poisoning defense (e.g. Geiping et al. [8] used it as a baseline defense). In this defense, adversarial inputs are generated on-the-fly using a known evasion attack so as to improve the model's generalization performance on the adversarial distribution. In our evaluation, we use the Projected Gradient Descent (PGD) attack, as is traditional, with the same hyperparemeters used by Geiping et al. [8] for consistency.

**Data Augmentation** techniques such as Mixup [27], Cutout [4], and CutMix [26] use synthetically created data to improve the model's generalization. Maxup [9] applies a set of these data augmentation techniques multiple times and selects the worst-case input for training to further improve generalization. As existing poisoning attacks relied on precise and sometimes large input manipulations, random data augmentation can introduce variability that the attacks are not prepared to address. Borgnia et al. [2] propose this approach, using Maxup with Cutout, as a poisoning defense.

## 3 Metrics to Evaluate Poisoning Defenses

**Security and Accuracy Metrics:** Traditionally, defenses are evaluated by measuring their performance in terms of the clean accuracy and final attack success rate. The clean accuracy (also known as benign accuracy) is the accuracy of the model evaluated on the test set with no poisoned samples. The attack success rate is the percentage of poisoned samples in the test set that were successfully misclassified. We use both of these metrics in our evaluations.

**Model Quality Metric:** In this paper, we introduce a new metric to help determine what effect various defenses have on different benign sub-populations in the dataset. Ideally, applying a defense should not result in a model that incorrectly predicts benign inputs coming from sub-populations. Samples in these sub-populations are typically uncommon or "difficult" to predict.

We utilize the statistical parity difference (SPD) [5] to determine how a particular defense treats benign samples from different sub-populations. Let us denote a feature set by $X$, the corresponding label set by $Y$ and the cardinality by $|\cdot|$. Given a class $y \in Y$, two populations within this class $P_1 := (X_1, Y_1)$ and $P_2 := (X_2, Y_2)$ consisting of solely *benign data samples* and a model $\mathcal{M} : X \rightarrow Y$ trained with a defense, the corresponding SPD for this given class $y \in Y$ can be computed as follows:

$$SPD = \frac{|\{(x,y) \in P_1 : \mathcal{M}(x) = y\}|}{|P_1|} - \frac{|\{(x,y) \in P_2 : \mathcal{M}(x) = y\}|}{|P_2|} \qquad (1)$$

We interpret the above metric based on the range of the value following standard conventions [1]:

- $|SPD| \leq 0.1$: An acceptable range, where none of the populations is disproportionately misclassified compared to the other. We refer to this as a fair outcome.

- $|SPD| > 0.1$: An unfair outcome where the evaluated model is biased towards $P_1$ (if $SPD > 0.1$) or $P_2$ (if $SPD < -0.1$).

Ideally, applying a defense should not disproportionately reduce the model performance for a particular benign population. Hence, an $|SPD| \leq 0.1$ is desirable. Clearly, applying a defense should not exacerbate the misclassification rate of these types of benign sub-populations.

There are multiple ways to identify benign subpopulations and some of them are dependent on the use case and dataset at hand. We highlight a method to identify this subpopulations in the following section, but note that there are many more ways to perform this task. One way to preform this task is to segment benign samples using *contextual data collection information* such as the time of the day when samples where collected or lightning conditions in a way that one benign subpopulation contains all daytime observations while the other night time samples. Other potential ways to segment samples include using sensitive attributes as defined in the fairness literature [1], e.g., younger vs. older, or by selecting samples that are not well represented in the training set.

## 4 Benchmarking of Poisoning Defenses

### 4.1 Experimental Setup

We evaluate the **seven** popular defenses presented in Section 2 under multiple poisoning attacks. We also use a modified combination defense where we apply both Data Augmentation and Adversarial Training (D.A. + A.T.), which is expected to mitigate the drop in clean accuracy when doing adversarial training alone. Specifically, we apply CutMix to augment the data and perform adversarial training on a fraction of the data; for our experiments we choose 75% as the hyperparameter. We test against a dirty-label backdoor attack [10] (DLBD) and a clean-label backdoor attack [22] (CLBD). Under both attacks, only samples from one of the classes (the target class) are poisoned. To fully understand the effect that an adversary may have over the model, we vary the percentage of poison for each defense. For DLBD, we use 0%, 1%, 5%, 10%, 20%, and 30% poison. For CLBD, we use 0%, 20%, 50%, and 80% poison. Note that this poison is only applied to the target class. In addition, for both types of attacks, we used two different triggers: a *bullet hole* and a *peace sign*. For more information on the triggers used in our experiments, refer to Appendix A.3. We present the results where the defenses are evaluated using the MicronNet model [25] trained on the GTSRB dataset [12]. All experiments were run using the Armory framework [20].

We only present the evaluations using the default hyperparameters for each defense as described in Section 2. For additional evaluations using different hyperparameters refer to Appendix A.4. To further test generalizability, we also evaluate these defenses using the CIFAR-10 dataset [14] against the DLBD [10] and Witches' Brew [7] attacks. Refer to Appendix A.5 for these results.

**Baselines:** We compare the performance of these defenses to a variety of baselines to understand their effect on the security and fairness metrics. As baselines, we use the following models.

1. *Undefended:* This defense mimics the scenario where a model is poisoned and no corrective measure is applied, creating a worst case scenario.

2. *Perfect filter:* This is an oracle filtering defense that fully removes the poisoned data from the training set. This is the best case scenario.

3. *Random filter:* This baseline randomly removes a percentage of the data, allowing us to determine how simply removing some data compares with popular defenses. We include this baseline given that our preliminary experimental results suggest that popular filtering defenses remove around 10% of the training data regardless of whether benign or poisonous.

**Sub-population Generation:** To determine different sub-populations within the benign training data, we make use of an explanatory model called BEAN regularization [6]. Then, given a test set, for each sample within a class, the explanatory model predicts whether the input sample belongs to a well-represented group (population 1) or out-of-distribution group (population 2). This labeling process is performed on each class in the dataset and fed to the SPD metric previously defined in (1). For more information on the BEAN model and why we choose this, refer to Appendix A.2.

## 4.2 Experimental Results

We now present the results for the **three** baselines and **eight** defenses we evaluated. Every baseline or defense, except Finite Aggregation, was evaluated for a total of three trials[1]. We report the average metrics among all trials.

**Security Assessment:** We first evaluate the clean accuracy and attack success rate for the defenses and baselines. In Figure 1, we report these metrics across the varying poisoning percentages for the two attacks using the bullet holes and peace sign triggers. We omit 0% poison for the attack success rate plots. We also include tables showing the exact numbers in Appendix A.6.

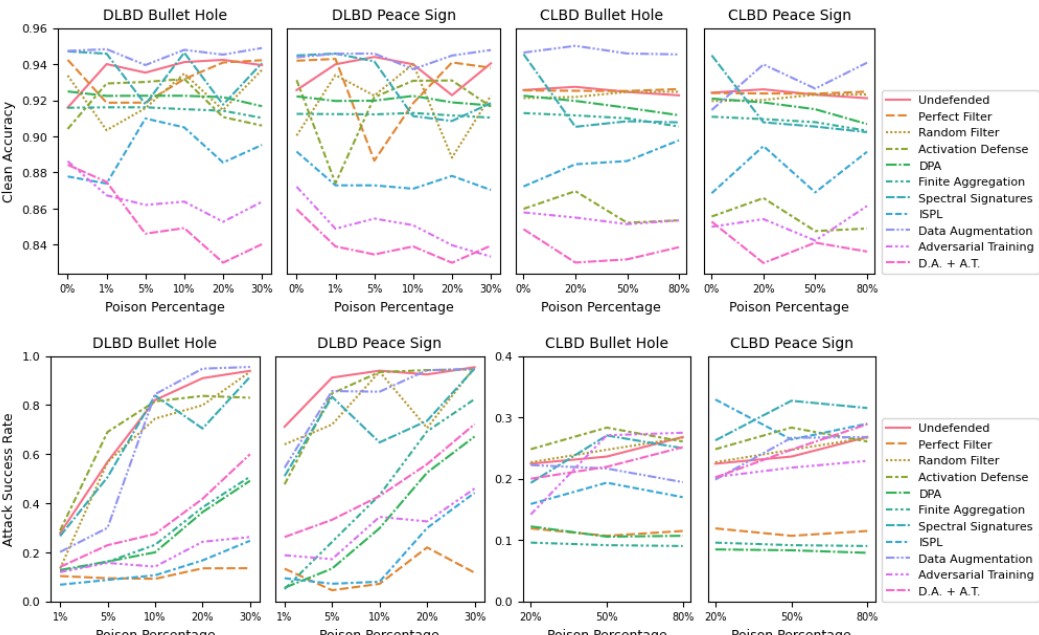

Figure 1: The clean accuracy (top) and attack success rate (bottom) for the baselines and defenses across varied poison. Each model is evaluated against the DLBD and CLBD attacks using the bullet hole and peace sign triggers on the GTSRB dataset.

We first notice that there is no clear correlation between poison percentage and clean accuracy for either attack type or trigger. We do observe, however, that most of the defenses have lower clean accuracy than the baselines. Particularly, Adversarial Training and the D.A. + A.T. combo tend to have the lowest clean accuracy. This is consistent with the fact that training on adversarial examples usually hurts the model performance [17]. Data Augmentation is an exception where it often performs

---

[1]For Finite Aggregation, we only ran one trial as it requires training a large number of models (e.g., with the default parameters of $k = 50$, $d = 10$, 500 models MicronNet models need to be trained), requiring 3 days to run a single experiment using a 32GB V100 GPU under GTSRB.

even better than the baselines. Most of the defenses have similar accuracy for both DLBD and CLBD; the exception is Activation Defense which performs worse against CLBD.

For the attack success rate, we observe that in DLBD, nearly all of the defenses are more robust than the Undefended and Random Filter baselines. This does not extend to CLBD as many of the defenses are actually less robust than the Undefended baseline. We also notice that the choice of trigger can affect some of the defenses such as ISPL for CLBD, with the peace sign being a more nefarious trigger. DPA and Finite Aggregation consistently offer very high levels of robustness for both attacks, and even outperform the Perfect Filter baseline during CLBD.

**Fairness Assessment:** We visualize the resulting SPD metric of the models using heatmaps by aggregating the fairness of classes across different models for scenarios with different poison percentages. To generate this heatmap, for each model and poison percentage, we first categorize the resulting SPD values associating with each class into SPD intervals described in Section 3. The classes belonging to $SPD < -0.1$ were considered as negative classes and $SPD > 0.1$, were considered as positive classes and $|SPD| \leq 0.1$ were considered as fair classes. The counts of classes belonging to each of these three categories were taken across different poison percentages and averaged for all trials. This was performed individually for all defenses or models under consideration and were plotted against each other. In Figure 2, we visualize the number of classes where $|SPD| \leq 0.1$ across the varying poisoning percentages and the percentage of classes for each of the three intervals averaged across all poison percentages for the two attacks using the bullet holes and peace sign triggers.

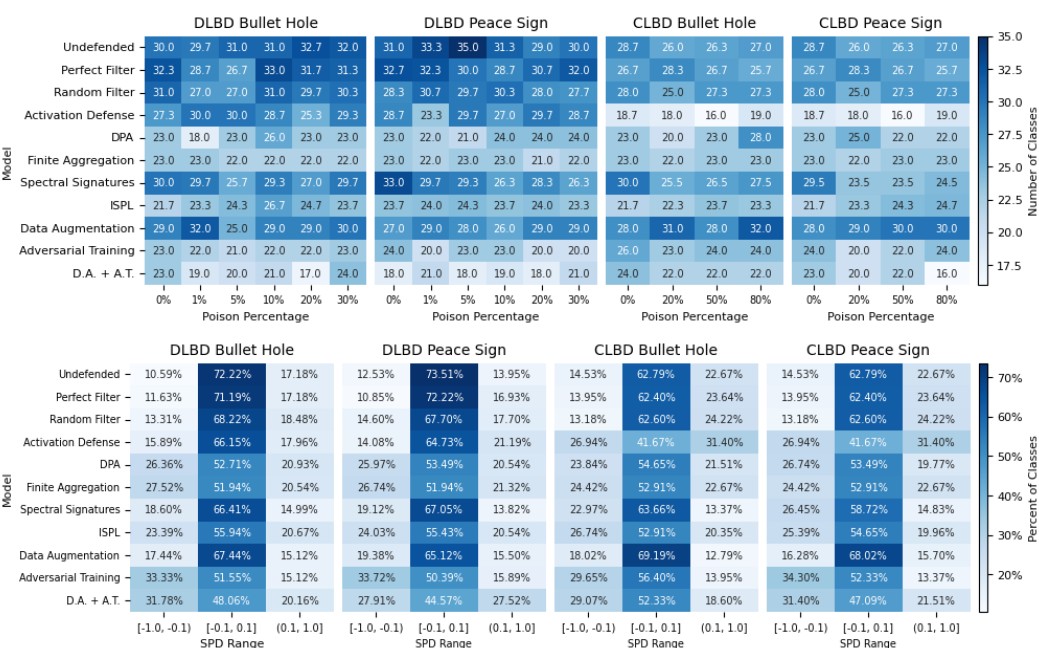

Figure 2: The number of classes within the acceptable $|SPD| \leq 0.1$ range across varied poison (top) and the percentage of classes within each of the three SPD ranges averaged across all poison percentages (bottom) for each baseline and defense. Each model is evaluated against the DLBD and CLBD attacks using the bullet hole and peace sign triggers on the GTSRB dataset.

In the heatmaps for the class counts of the $|SPD| \leq 0.1$ (top row of Figure 2), *darker blue* reflects that more classes fall into the acceptable range, and thus, the model is fairer over all. The different baselines are also shown for relative comparison purposes in each row. We first notice that the poison percentage does not have a large effect on the fairness as the class counts stay relatively consistent. The baseline models are the most fair while the evaluated defenses tend to be less fair. Interestingly, Activation Defense exhibits high levels of fairness during DLBD but the lowest fairness during CLBD. However, we observe that all of the filtering and adversarial training-based defenses are always less fair than the baselines and other defenses. Intuitively, this reflects the mechanics of the defenses, which remove outliers or try to minimize the effect of minority data. In contrast, data Augmentation consistently shows the highest fairness for both types of attacks.

In the heatmaps for the class percentages of the three intervals (bottom row of Figure 2), we notice that most defenses are not especially unfair as the majority of classes always lie within the $|SPD| \leq 0.1$ range. The exceptions are Activation Defense and the D.A. + A.T. combo which in some cases drop below 50% indicating they may be biased. We also observe that each baseline or defense tends to be skewed towards the either the negative $SPD < -0.1$ or positive $SPD > 0.1$. The baselines and Activation Defense tend to be skewed towards the positive range with the percentage of positive range larger that the negative range, while the other defenses tend to be skewed towards negative range. This pattern is consistent for both DLBD and CLBD and for both triggers. This shows that most of the defenses are biased towards a specific sub-population regardless of the type of poisoning attacks.

## 5    Conclusion

Machine learning algorithms, although used for critical tasks, are susceptible to adversarial attacks. Poisoning attacks, in particular, pose a large risk to these ML models. Many defenses have been proposed to protect against poisoning attacks. Traditionally, these defenses have been evaluated using attack success rate and benign accuracy. However, these metrics do not show the complete way in which a defense may influence the model. To uncover potential side-effects of defenses, we introduced the use of a fairness metric to understand how different sub-populations can be affected.

In our evaluations, we found that some defenses that produce robust models with a low attack success rate can actually yield unfair and biased models with a low amount of classes in the acceptable Statistical Parity Difference (SPD) range. This was particularly true for the filtering and adversarial training defenses. Overall, our work highlights that creating robust models may have unintended consequences on the final model quality and certain sub-populations. We encourage future evaluations of adversarial defenses to use metrics outside of the traditional clean accuracy and attack success rate such as SPD to measure additional qualities of the model including its resulting fairness.

## Acknowledgement

The work contained herein is developed under the DARPA GARD program. This material is based upon work supported by the Defense Advanced Research Projects Agency (DARPA) under Contracts No. HR001120C0013., No. HR001120C0114 and Basic Contract No. W56KGU-18-D-0004. Any opinions, findings and conclusions or recommendations expressed in this material are those of the author(s) and do not necessarily reflect the views of the Defense Advanced Research Projects Agency (DARPA). The views, opinions and/or findings contained in this report are those of The MITRE Corporation, IBM and Two Six Technologies and should not be construed as an official government position, policy, or decision, unless designated by other documentation. Approved for Public Release by MITRE. Distribution Unlimited. Public Release Case Number 22-3446. ©2022 IBM and The MITRE Corporation. All rights reserved.

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

# A    Appendix

### A.1    Sub-population Examples

Many adversarial poisoning defenses attempt to find poisoned samples by searching for sub-populations that are different from the rest of the samples in the class. However, it is possible for naturally occurring low-represented benign sub-populations to be falsely identified in this process. In different samples, we observed difference in properties like brightness, contrast, blurriness, zoom-in or zoom-out etc., can create a sub-population that may be potentially marked as poison. One such example is Figure 3 where the difference in brightness can create a sub-population that may potentially be marked as poison.

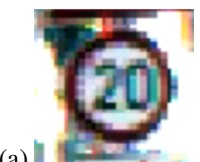 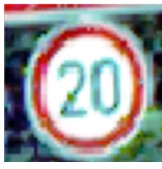 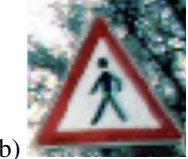 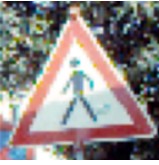

(a)                    (b)

Figure 3: Sample images in the GTSRB dataset where sub-populations can naturally occur. Here, the (a) speed limit and (b) pedestrian crossing signs appear to be different sub-populations due to differences in brightness.

### A.2    BEAN Regularization to Identify Sub-populations

As explained before, there are multiple ways to identify benign sub-populations. In our experiments, we used the BEAN regularization explanatory model to determine the different sub-populations within the training data. BEAN allows for semantic interpretability due to its layer-wise regularization rules that are biologically motivated. By applying these learning constraints which adjust the weight space (enforcing modularity) BEAN ultimately allows sparsifying connections to disentangle learned concepts into distinct groups. Our qualitative tests reproduced this same behavior on multiple datasets (including GTSRB and CIFAR-10) and architectures. While the original paper tests BEAN for better generalization and zero-shot learning, in this study, we leveraged the trait of explanatory data characterization in detecting out-of-distribution sub-populations within a single "learned" class of the pretrained BEAN model. The pretrained BEAN model uses the same architecture as the undefended model, but solely for the purpose of evaluating the defense model behavior on non-poisoned inputs only.

### A.3 Trigger Images

In our experiments, we evaluated the baselines and defenses against the dirty-label and clean-label backdoor attacks using the bullet hole and peace sign triggers for the GTSRB dataset. We also ran additional experiments on the CIFAR-10 dataset using the copyright and watermark triggers. We generate poison samples by blending the trigger image with a portion of each target image [2]. For GTSRB, we position the trigger in the center of the image and use a blend factor of 0.6 for both triggers. For CIFAR-10, we set the trigger image size equal to the original image and use a blend factor of 0.18 for copyright and 1.0 for the watermark trigger. All of these are shown in Figure 4.

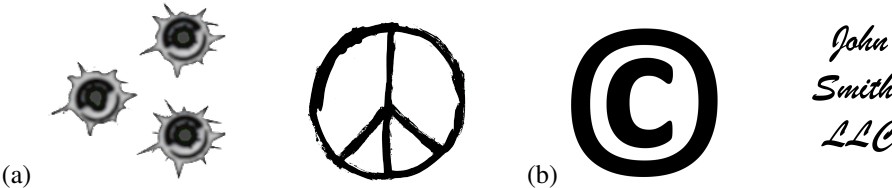

(a)                                                                            (b)

Figure 4: Poisoning attack triggers for (a) the bullet hole and peace sign used for the GTSRB dataset and (b) the copyright and watermark used for the CIFAR-10 dataset.

### A.4 Alternative Hyperparameter Evaluations

In our evaluations, we used the default hyperparameters for all of the defenses based on their respective paper. We now present results for the Activation Defense, Spectral Signatures, Data Augmentation, Adversarial Training, and the Data Augmentation and Adversarial combination (D.A. + A.T.) defenses where we use alternative hyperparameters. For Activation Defense, there is an optional exclusionary reconstruction phase [3] (we denote as this as ExRe); we now evaluate using exclusionary reconstruction with a threshold of 1.0. For Spectral Signatures, we evaluated using the best case scenario where the expected poison is the actual poison amount; now we use a fixed 30% poison for the dirty label attacks and fixed 20% poison for the clean label attacks. For Data Augmentation, Borgnia et al. [2] used Maxup with Cutout; now we use the CutMix, a different data augmentation method. For Adversarial Training, Geiping et al. [8] used a 7-step PGD attack with $\epsilon = 0.1$ and a step size of 0.02; we now use a weakened PGD attack with the same hyperparamaters that Madry et al. [17] used, a 10-step PGD with $\epsilon = 0.03$ and a step size of 0.007. For the D.A. + A.T. combo, we originally adversarially perturbed 75% of the training samples; we now use a weakened version where we only perturb 50% of the training samples.

**Security Assessment:** The clean accuracy and attack success rate for these alternative hyperparameter defenses are shown in Figure 5. The undefended baseline and original hyperparameter variations are also included as reference. We report these metrics across the varying poisoning percentages for the two attack using the bullet holes and peace sign triggers. We omit 0% poison for the attack success rate plots since there will never be a successful attack. We also include tables showing the exact numbers in Appendix A.6. We immediately notice that Activation Defense with exclusionary reconstruction offers a significant improvement in clean accuracy compared to without but does not change much in terms of attack success rate. We also observe that the choice of expected poison for Spectral Signatures does not affect the clean accuracy but does have a significant effect on the attack success rate. The choice of Data Augmentation method (Maxup or CutMix) does not have a very large affect on the clean accuracy or attack success rate. For both Adversarial Training and the D.A. + A.T., the weakened variant has a higher clean accuracy but also higher attack success rate.

**Fairness Assessment:** In Figure 6, we report the number of classes in the acceptable $|SPD| \leq 0.1$ across the varying poisoning percentages and the percentage of classes for each of the three SPD intervals averaged across all poison percentages for the two attacks using the bullet holes and peace sign triggers. We immediately notice a large improvement in fairness for Activation Defense with exclusionary reconstruction for CLBD. We also observe that the hyperparameters for Spectral Signatures and the choice of Data Augmentation do not affect the fairness of the model as the SPD remains very similar. For both Adversarial Training and the D.A. + A.T. combo, however, we notice

---

[2]Refer to `https://github.com/Trusted-AI/adversarial-robustness-toolbox/blob/main/art/attacks/poisoning/perturbations/image_perturbations.py` for the implementation details

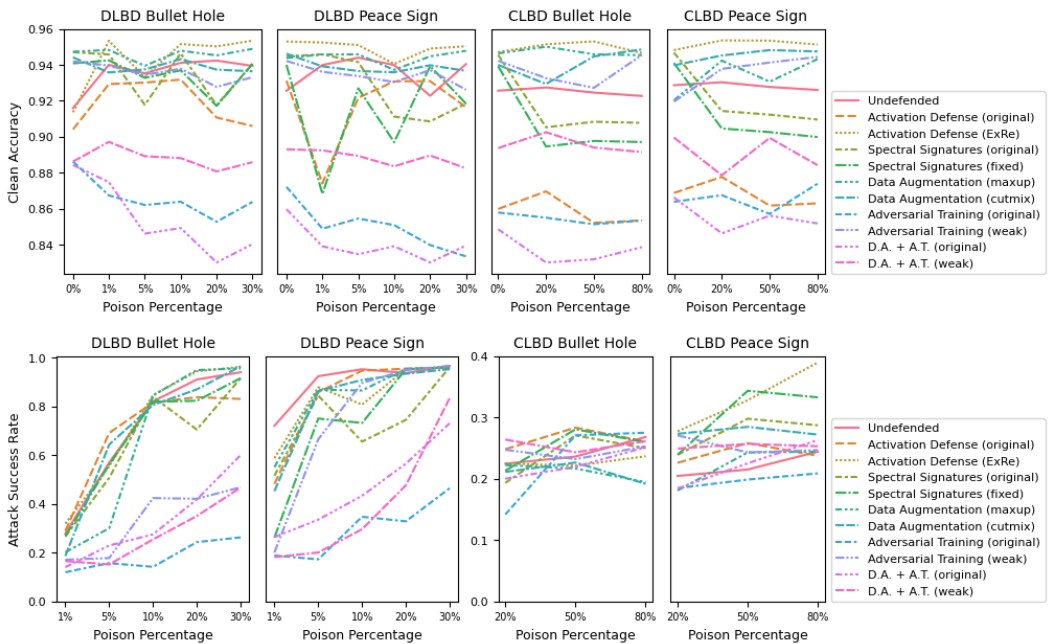

Figure 5: The clean accuracy (top) and attack success rate (bottom) for the additional hyperparameter modified defenses across varied poison. Each model is evaluated against the DLBD and CLBD attacks using the bullet hole and peace sign triggers on the GTSRB dataset.

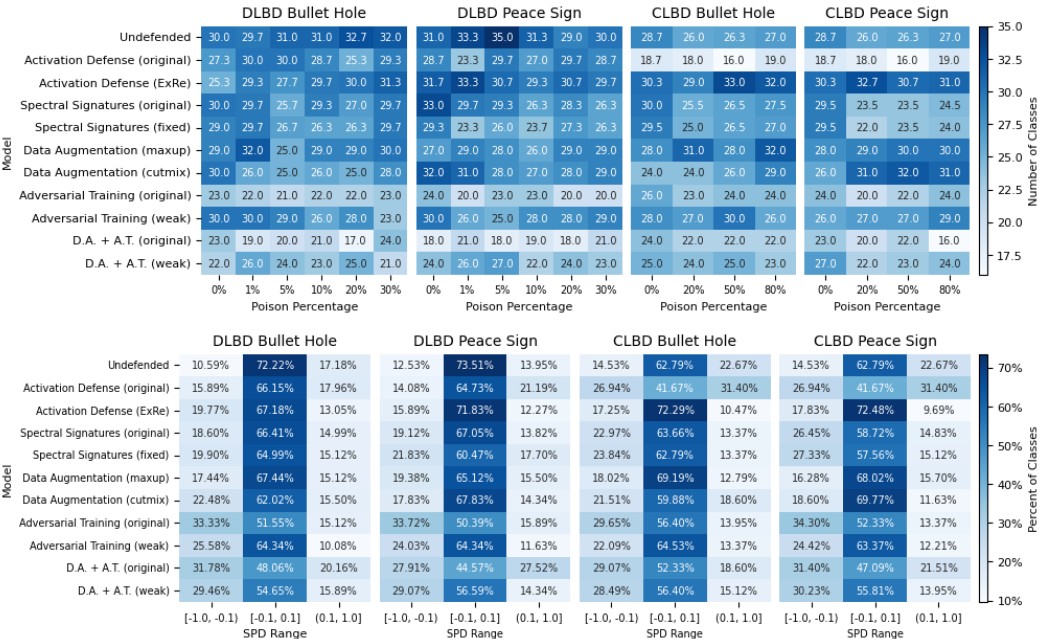

Figure 6: The number of classes within the acceptable $|SPD| \leq 0.1$ range across varied poison (top) and the percentage of classes within each of the three SPD ranges averaged across all poison percentages (bottom) for the additional hyperparameter modified defenses. Each model is evaluated against the DLBD and CLBD attacks using the bullet hole and peace sign triggers on the GTSRB dataset.

a large increase in fairness for the weakened variants. The same trend where the defense tends to be skewed towards the either the negative $SPD < -0.1$ or positive $SPD > 0.1$ remains consistent regardless of the hyperparameters.

## A.5   Alternative Dataset and Attack Evaluations

In addition to our experiments using the MicronNet model trained on the GTSRB dataset, we also use a different model architecture and dataset to properly judge each defense's generalizability. For each of the baselines and defenses, except DPA and Finite Aggregation [3], we trained a ResNet-18 [11] on the CIFAR-10 [14]. Each model was trained for 100 epochs. We evaluate these defenses against the dirty label backdoor [10] and Witches' Brew [7] attacks. We use the Adam optimizer for models evaluated against the DLBD and the SGD optimizer for models evaluated against the Witches' Brew attack. The CIFAR-10 dataset uses the copyright and watermark triggers (see Appendix A.3) for the DLBD. The Witches' Brew attack does not use a trigger and we use 0%, 1%, 5%, 10%, 20%, and 30% poison.

**Security Assessment:** In Figure 7, we report the clean accuracy and attack success rate across the varying poisoning percentages for the two attacks (and triggers where applicable). We omit 0% poison for the attack success rate plots since there will never be a successful attack. We also include tables showing the exact numbers in Appendix A.6. We immediately notice that nearly all of the defenses have a lower clean accuracy than the baselines. Adversarial Training and the D.A. + A.T. combo in particular perform extremely poorly, but this may be due to the choice of hyperparameters for the PGD attack. For the attack success rate, the defenses perform better than the undefended baseline for the DLBD. However, for the Witches' Brew attack, all of the defenses perform worse than the baseline. This may demonstrate that these defenses are brittle against this type of attack.

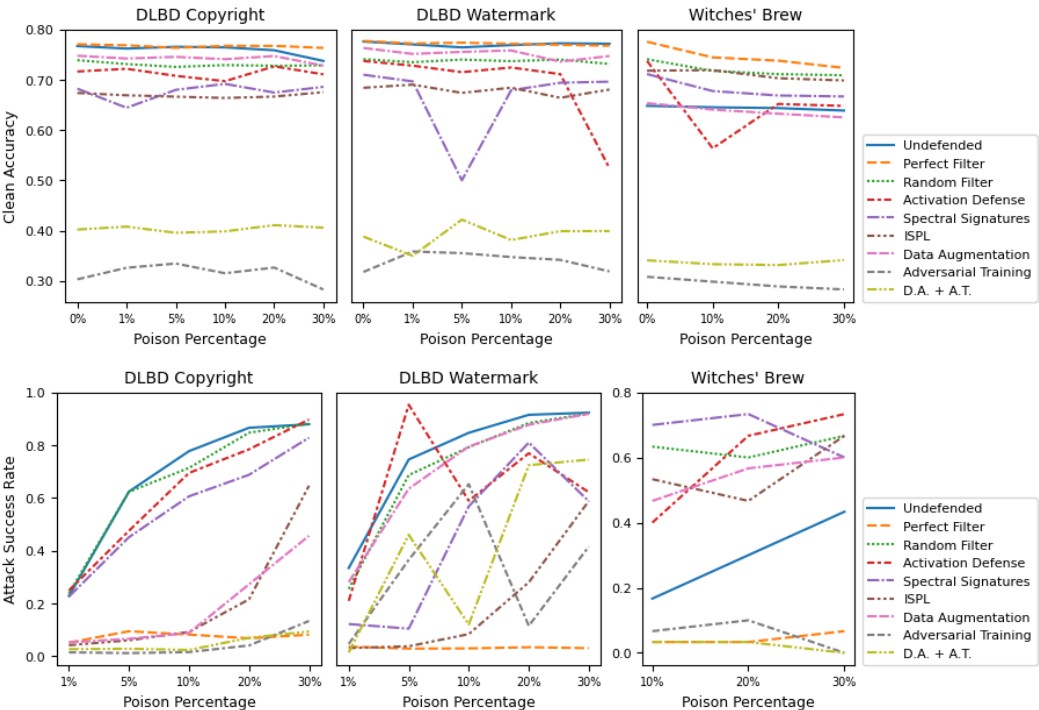

Figure 7: The clean accuracy (top) and attack success rate (bottom) for the baselines and defenses across varied poison. Each model is evaluated against the DLBD attack using the copyright and watermark triggers and the Witches' Brew attack on the CIFAR-10 dataset.

**Fairness Assessment:** The fairness class counts for the acceptable $|SPD| \leq 0.1$ range and the percent distribution for all three SPD ranges are shown in Figure 8. We observe that since the

---

[3]We omit DPA and Finite Aggregation for the CIFAR-10 results as these defenses were not compatible with this combination of dataset and poisoning attacks.

CIFAR-10 dataset tends to have balanced classes, the baselines and most of the defenses are very fair and tend to exhibit no bias. Adversarial Training and the D.A. + A.T. combo show some form of bias, but is still very low. This shows that against a balanced dataset, models tend to stay fair and do not exhibit much bias against sub-populations. For this reason, we used the GTSRB dataset for our main results as this dataset is unbalanced which is more realistic.

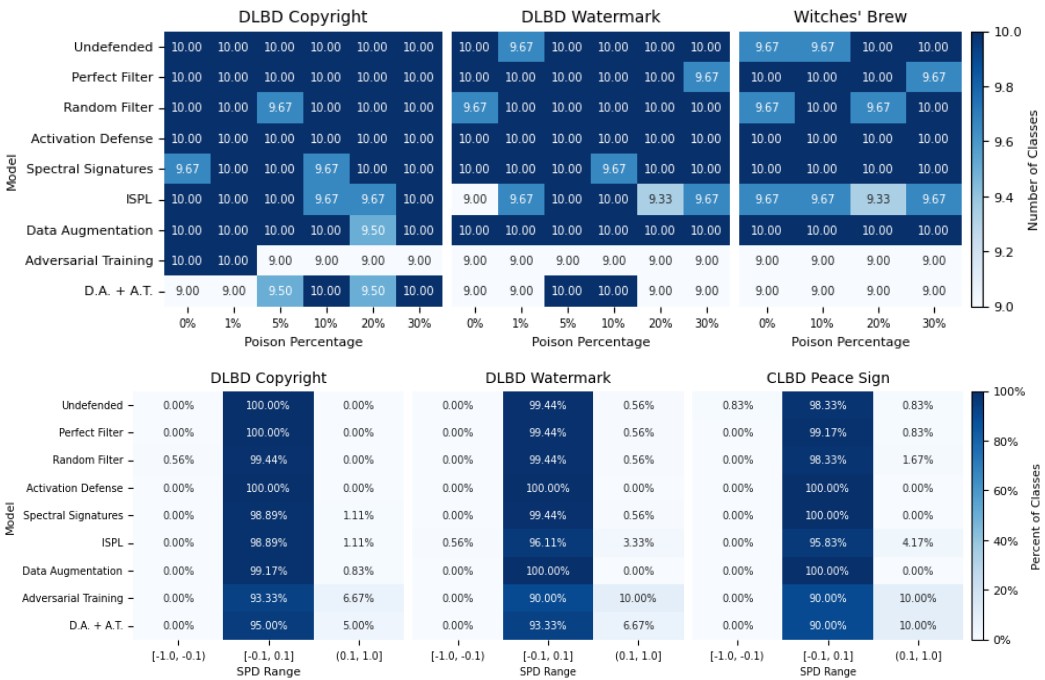

Figure 8: The number of classes in the acceptable $|SPD| \leq 0.1$ range across varied poison (top) and the percentage of classes within each of the three SPD ranges averaged across all poison percentages (bottom) for each baseline and defense. Each model is evaluated against the DLBD attack using the copyright and watermark triggers and the Witches' Brew attack on the CIFAR-10 dataset.

## A.6 Results as Tables

For readability and accurate comparisons, we also include tables showing the exact numbers for the clean accuracy and attack success rate plots of all the baselines and defenses in our three sets of experiments.

Table 1 and Table 2 correspond to the top row and bottom rows of Figure 1, respectively. This is the clean accuracy and attack success rates for the main set of baselines and defenses evaluated against the DLBD and CLBD attacks using the bullet hole and peace sign triggers on the GTSRB dataset.

Table 3 and Table 4 correspond to the top row and bottom rows of Figure 5, respectively. This is the clean accuracy and attack success rates for the alternative hyperparameter defenses evaluated against the DLBD and CLBD attacks using the bullet hole and peace sign triggers on the GTSRB dataset.

Table 5 and Table 6 correspond to the top row and bottom rows of Figure 7, respectively. This is the clean accuracy and attack success rates for the alternative hyperparameter defenses evaluated against the DLBD attack using the copyright and watermark triggers and Witches' Brew attack on the CIFAR-10 dataset.

Table 1: The clean accuracy for the main set of baselines and defenses evaluated on the GTSRB dataset against the DLBD (top) and CLBD (bottom) attacks with bullet hole and peace sign triggers across varying poison percentages. The best performing models are in bold.

| Model | DLBD Bullet Hole | | | | | | DLBD Peace Sign | | | | | |
|---|---|---|---|---|---|---|---|---|---|---|---|---|
| | 0% | 1% | 5% | 10% | 20% | 30% | 0% | 1% | 5% | 10% | 20% | 30% |
| Undefended | 0.916 | 0.940 | 0.935 | 0.941 | 0.942 | 0.939 | 0.928 | 0.941 | 0.945 | 0.941 | 0.925 | 0.942 |
| Perfect Filter | 0.942 | 0.919 | 0.919 | 0.932 | 0.941 | 0.942 | 0.943 | 0.944 | 0.892 | 0.921 | 0.942 | 0.940 |
| Random Filter | 0.934 | 0.903 | 0.916 | 0.935 | 0.915 | 0.937 | 0.905 | 0.936 | 0.925 | 0.941 | 0.893 | 0.925 |
| Activation Defense | 0.904 | 0.929 | 0.930 | 0.932 | 0.911 | 0.906 | 0.933 | 0.880 | 0.924 | 0.933 | 0.933 | 0.920 |
| DPA | 0.925 | 0.922 | 0.922 | 0.922 | 0.922 | 0.917 | 0.925 | 0.922 | 0.923 | 0.925 | 0.922 | 0.920 |
| Finite Aggregation | 0.916 | 0.916 | 0.916 | 0.915 | 0.914 | 0.910 | 0.916 | 0.916 | 0.916 | 0.916 | 0.915 | 0.914 |
| Spectral Signatures | 0.947 | 0.946 | 0.918 | 0.946 | 0.918 | 0.941 | **0.946** | 0.947 | 0.943 | 0.915 | 0.912 | 0.921 |
| ISPL | 0.878 | 0.874 | 0.910 | 0.905 | 0.885 | 0.895 | 0.896 | 0.879 | 0.879 | 0.877 | 0.884 | 0.877 |
| Data Augmentation | **0.947** | **0.948** | **0.939** | **0.948** | **0.945** | **0.949** | 0.945 | **0.947** | **0.947** | **0.939** | **0.946** | **0.949** |
| Adversarial Training | 0.886 | 0.867 | 0.862 | 0.864 | 0.853 | 0.864 | 0.878 | 0.857 | 0.862 | 0.859 | 0.848 | 0.842 |
| D.A. + A.T. | 0.884 | 0.875 | 0.846 | 0.849 | 0.830 | 0.840 | 0.867 | 0.848 | 0.844 | 0.848 | 0.839 | 0.848 |

| Model | CLBD Bullet Hole | | | | CLBD Peace Sign | | | |
|---|---|---|---|---|---|---|---|---|
| | 0% | 20% | 50% | 80% | 0% | 20% | 50% | 80% |
| Undefended | 0.928 | 0.929 | 0.927 | 0.925 | 0.928 | 0.929 | 0.927 | 0.925 |
| Perfect Filter | 0.928 | 0.927 | 0.927 | 0.928 | 0.928 | 0.927 | 0.927 | 0.928 |
| Random Filter | 0.924 | 0.924 | 0.927 | 0.927 | 0.924 | 0.924 | 0.927 | 0.927 |
| Activation Defense | 0.866 | 0.875 | 0.859 | 0.860 | 0.866 | 0.875 | 0.859 | 0.860 |
| DPA | 0.925 | 0.922 | 0.919 | 0.915 | 0.925 | 0.923 | 0.920 | 0.912 |
| Finite Aggregation | 0.916 | 0.915 | 0.913 | 0.909 | 0.916 | 0.915 | 0.913 | 0.909 |
| Spectral Signatures | 0.946 | 0.909 | 0.912 | 0.911 | **0.946** | 0.913 | 0.911 | 0.908 |
| ISPL | 0.878 | 0.889 | 0.891 | 0.902 | 0.878 | 0.901 | 0.878 | 0.898 |
| Data Augmentation | **0.947** | **0.951** | **0.947** | **0.946** | 0.919 | **0.942** | 0.930 | **0.943** |
| Adversarial Training | 0.864 | 0.862 | 0.858 | 0.860 | 0.861 | 0.865 | 0.854 | 0.871 |
| D.A. + A.T. | 0.856 | 0.838 | 0.840 | 0.846 | 0.864 | 0.843 | 0.853 | 0.849 |

Table 2: The attack success rate for the main set of baselines and defenses evaluated on the GTSRB dataset against the DLBD (top) and CLBD (bottom) attacks with the bullet hole and peace sign triggers across varying poison percentages. The best performing models are in bold.

| Model | DLBD Bullet Hole | | | | | DLBD Peace Sign | | | | |
|---|---|---|---|---|---|---|---|---|---|---|
| | 1% | 5% | 10% | 20% | 30% | 1% | 5% | 10% | 20% | 30% |
| Undefended | 0.278 | 0.569 | 0.820 | 0.910 | 0.940 | 0.744 | 0.953 | 0.983 | 0.967 | 0.998 |
| Perfect Filter | 0.103 | 0.094 | **0.092** | **0.135** | **0.135** | 0.137 | **0.045** | **0.072** | **0.228** | **0.120** |
| Random Filter | 0.134 | 0.568 | 0.745 | 0.800 | 0.936 | 0.668 | 0.753 | 0.979 | 0.738 | 0.998 |
| Activation Defense | 0.289 | 0.691 | 0.815 | 0.838 | 0.830 | 0.497 | 0.886 | 0.977 | 0.985 | 0.988 |
| DPA | 0.128 | 0.163 | 0.200 | 0.364 | 0.492 | 0.056 | 0.138 | 0.310 | 0.547 | 0.701 |
| Finite Aggregation | 0.122 | 0.161 | 0.231 | 0.383 | 0.508 | **0.050** | 0.251 | 0.451 | 0.724 | 0.861 |
| Spectral Signatures | 0.266 | 0.506 | 0.839 | 0.705 | 0.915 | 0.532 | 0.871 | 0.676 | 0.769 | 0.992 |
| ISPL | **0.068** | **0.088** | 0.106 | 0.167 | 0.247 | 0.096 | 0.073 | 0.081 | 0.311 | 0.461 |
| Data Augmentation | 0.201 | 0.300 | 0.844 | 0.949 | 0.956 | 0.568 | 0.896 | 0.893 | 0.985 | 0.990 |
| Adversarial Training | 0.119 | 0.157 | 0.142 | 0.243 | 0.263 | 0.194 | 0.178 | 0.358 | 0.339 | 0.479 |
| D.A. + A.T. | 0.140 | 0.229 | 0.275 | 0.418 | 0.600 | 0.272 | 0.346 | 0.447 | 0.583 | 0.754 |

| Model | CLBD Bullet Hole | | | CLBD Peace Sign | | |
|---|---|---|---|---|---|---|
| | 20% | 50% | 80% | 20% | 50% | 80% |
| Undefended | 0.225 | 0.236 | 0.268 | 0.225 | 0.236 | 0.268 |
| Perfect Filter | 0.119 | 0.107 | 0.115 | 0.119 | 0.107 | 0.115 |
| Random Filter | 0.227 | 0.247 | 0.267 | 0.227 | 0.247 | 0.267 |
| Activation Defense | 0.248 | 0.283 | 0.261 | 0.248 | 0.283 | 0.261 |
| DPA | 0.122 | 0.106 | 0.107 | **0.085** | **0.083** | **0.079** |
| Finite Aggregation | **0.096** | **0.092** | **0.090** | 0.096 | 0.092 | 0.090 |
| Spectral Signatures | 0.193 | 0.271 | 0.250 | 0.263 | 0.327 | 0.315 |
| ISPL | 0.159 | 0.194 | 0.170 | 0.329 | 0.264 | 0.290 |
| Data Augmentation | 0.222 | 0.217 | 0.194 | 0.199 | 0.267 | 0.268 |
| Adversarial Training | 0.142 | 0.271 | 0.275 | 0.203 | 0.218 | 0.229 |
| D.A. + A.T. | 0.200 | 0.219 | 0.251 | 0.202 | 0.247 | 0.289 |

Table 3: The clean accuracy for the alternative hyperparameter defenses evaluated on the GTSRB dataset against the DLBD (top) and CLBD (bottom) attacks with bullet hole and peace sign triggers across varying poison percentages. The best performing models are in bold.

| Model | DLBD Bullet Hole | | | | | | DLBD Peace Sign | | | | | |
|---|---|---|---|---|---|---|---|---|---|---|---|---|
| | 0% | 1% | 5% | 10% | 20% | 30% | 0% | 1% | 5% | 10% | 20% | 30% |
| Undefended | 0.916 | 0.940 | 0.935 | 0.941 | 0.942 | 0.939 | 0.928 | 0.941 | 0.945 | 0.941 | 0.925 | 0.942 |
| Activation Defense (original) | 0.904 | 0.929 | 0.930 | 0.932 | 0.911 | 0.906 | 0.933 | 0.880 | 0.924 | 0.933 | 0.933 | 0.920 |
| Activation Defense (ExRe) | 0.914 | **0.953** | 0.933 | **0.952** | **0.950** | **0.953** | **0.954** | **0.953** | **0.952** | **0.942** | **0.950** | **0.951** |
| Spectral Signatures (original) | 0.947 | 0.946 | 0.918 | 0.946 | 0.918 | 0.941 | 0.946 | 0.947 | 0.943 | 0.915 | 0.912 | 0.921 |
| Spectral Signatures (fixed) | 0.941 | 0.943 | 0.933 | 0.937 | 0.917 | 0.941 | 0.941 | 0.875 | 0.929 | 0.901 | 0.941 | 0.921 |
| Data Augmentation (maxup) | **0.947** | 0.948 | **0.939** | 0.948 | 0.945 | 0.949 | 0.945 | 0.947 | 0.947 | 0.939 | 0.946 | 0.949 |
| Data Augmentation (cutmix) | 0.944 | 0.936 | 0.937 | 0.943 | 0.937 | 0.936 | 0.947 | 0.941 | 0.938 | 0.938 | 0.941 | 0.938 |
| Adversarial Training (original) | 0.886 | 0.867 | 0.862 | 0.864 | 0.853 | 0.864 | 0.878 | 0.857 | 0.862 | 0.859 | 0.848 | 0.842 |
| Adversarial Training (weak) | 0.942 | 0.940 | 0.934 | 0.938 | 0.928 | 0.933 | 0.943 | 0.938 | 0.936 | 0.933 | 0.939 | 0.929 |
| D.A. + A.T. (original) | 0.884 | 0.875 | 0.846 | 0.849 | 0.830 | 0.840 | 0.867 | 0.848 | 0.844 | 0.848 | 0.839 | 0.848 |
| D.A. + A.T. (weak) | 0.886 | 0.897 | 0.889 | 0.888 | 0.881 | 0.886 | 0.898 | 0.897 | 0.894 | 0.889 | 0.895 | 0.888 |

| Model | CLBD Bullet Hole | | | | CLBD Peace Sign | | | |
|---|---|---|---|---|---|---|---|---|
| | 0% | 20% | 50% | 80% | 0% | 20% | 50% | 80% |
| Undefended | 0.928 | 0.929 | 0.927 | 0.925 | 0.928 | 0.929 | 0.927 | 0.925 |
| Activation Defense (original) | 0.866 | 0.875 | 0.859 | 0.860 | 0.866 | 0.875 | 0.859 | 0.860 |
| Activation Defense (ExRe) | 0.948 | **0.952** | **0.954** | 0.947 | **0.948** | **0.953** | **0.953** | **0.951** |
| Spectral Signatures (original) | 0.946 | 0.909 | 0.912 | 0.911 | 0.946 | 0.913 | 0.911 | 0.908 |
| Spectral Signatures (fixed) | 0.940 | 0.899 | 0.902 | 0.901 | 0.940 | 0.903 | 0.901 | 0.898 |
| Data Augmentation (maxup) | **0.947** | 0.951 | 0.947 | 0.946 | 0.919 | 0.942 | 0.930 | 0.943 |
| Data Augmentation (cutmix) | 0.941 | 0.931 | 0.946 | **0.949** | 0.939 | 0.945 | 0.948 | 0.947 |
| Adversarial Training (original) | 0.864 | 0.862 | 0.858 | 0.860 | 0.861 | 0.865 | 0.854 | 0.871 |
| Adversarial Training (weak) | 0.943 | 0.934 | 0.929 | 0.946 | 0.918 | 0.937 | 0.941 | 0.944 |
| D.A. + A.T. (original) | 0.856 | 0.838 | 0.840 | 0.846 | 0.864 | 0.843 | 0.853 | 0.849 |
| D.A. + A.T. (weak) | 0.898 | 0.906 | 0.898 | 0.896 | 0.898 | 0.876 | 0.898 | 0.882 |

Table 4: The attack success rate for the alternative hyperparameter defenses evaluated on the GTSRB dataset against the DLBD (top) and CLBD (bottom) attacks with the bullet hole and peace sign triggers across varying poison percentages. The best performing models are in bold.

| Model | DLBD Bullet Hole | | | | | DLBD Peace Sign | | | | |
|---|---|---|---|---|---|---|---|---|---|---|
| | 1% | 5% | 10% | 20% | 30% | 1% | 5% | 10% | 20% | 30% |
| Undefended | 0.278 | 0.569 | 0.820 | 0.910 | 0.940 | 0.744 | 0.953 | 0.983 | 0.967 | 0.998 |
| Activation Defense (original) | 0.289 | 0.691 | 0.815 | 0.838 | 0.830 | 0.497 | 0.886 | 0.977 | 0.985 | 0.988 |
| Activation Defense (ExRe) | 0.319 | 0.554 | 0.846 | 0.944 | 0.962 | 0.606 | 0.906 | 0.833 | 0.983 | 0.992 |
| Spectral Signatures (original) | 0.266 | 0.506 | 0.839 | 0.705 | 0.915 | 0.532 | 0.871 | 0.676 | 0.769 | 0.992 |
| Spectral Signatures (fixed) | 0.268 | 0.562 | 0.819 | 0.823 | 0.915 | 0.270 | 0.774 | 0.756 | 0.983 | 0.992 |
| Data Augmentation (maxup) | 0.201 | 0.300 | 0.844 | 0.949 | 0.956 | 0.568 | 0.896 | 0.893 | 0.985 | 0.990 |
| Data Augmentation (cutmix) | 0.186 | 0.643 | 0.806 | 0.869 | 0.964 | 0.467 | 0.890 | 0.938 | 0.964 | 0.983 |
| Adversarial Training (original) | **0.119** | 0.157 | **0.142** | **0.243** | **0.263** | 0.194 | **0.178** | 0.358 | **0.339** | **0.479** |
| Adversarial Training (weak) | 0.171 | 0.178 | 0.424 | 0.421 | 0.468 | 0.207 | 0.686 | 0.925 | 0.979 | 0.997 |
| D.A. + A.T. (original) | 0.140 | 0.229 | 0.275 | 0.418 | 0.600 | 0.272 | 0.346 | 0.447 | 0.583 | 0.754 |
| D.A. + A.T. (weak) | 0.165 | **0.151** | 0.254 | 0.350 | 0.467 | **0.188** | 0.207 | **0.306** | 0.492 | 0.861 |

| Model | CLBD Bullet Hole | | | CLBD Peace Sign | | |
|---|---|---|---|---|---|---|
| | 20% | 50% | 80% | 20% | 50% | 80% |
| Undefended | 0.225 | 0.236 | 0.268 | 0.225 | 0.236 | 0.268 |
| Activation Defense (original) | 0.248 | 0.283 | 0.261 | 0.248 | 0.283 | 0.261 |
| Activation Defense (ExRe) | 0.225 | 0.221 | 0.237 | 0.305 | 0.360 | 0.428 |
| Spectral Signatures (original) | 0.193 | 0.271 | 0.250 | 0.263 | 0.327 | 0.315 |
| Spectral Signatures (fixed) | 0.213 | 0.281 | 0.260 | 0.263 | 0.377 | 0.365 |
| Data Augmentation (maxup) | 0.222 | **0.217** | 0.194 | **0.199** | 0.267 | 0.268 |
| Data Augmentation (cutmix) | 0.211 | 0.226 | **0.192** | 0.300 | 0.312 | 0.299 |
| Adversarial Training (original) | **0.142** | 0.271 | 0.275 | 0.203 | **0.218** | **0.229** |
| Adversarial Training (weak) | 0.247 | 0.232 | 0.253 | 0.297 | 0.267 | 0.271 |
| D.A. + A.T. (original) | 0.200 | 0.219 | 0.251 | 0.202 | 0.247 | 0.289 |
| D.A. + A.T. (weak) | 0.264 | 0.243 | 0.261 | 0.274 | 0.282 | 0.278 |

Table 5: The clean-accuracy for the baselines and defenses evaluated on the CIFAR-10 dataset against the DLBD attack with the copyright and watermark triggers (top) and Witches' Brew attack (bottom) across varying poison percentages. The best performing models are in bold.

| Model | DLBD Copyright | | | | | | DLBD Watermark | | | | | |
|---|---|---|---|---|---|---|---|---|---|---|---|---|
| | 0% | 1% | 5% | 10% | 20% | 30% | 0% | 1% | 5% | 10% | 20% | 30% |
| Undefended | 0.767 | 0.762 | **0.765** | 0.764 | 0.758 | 0.737 | 0.769 | 0.764 | 0.758 | 0.762 | **0.766** | **0.765** |
| Perfect Filter | **0.770** | **0.768** | 0.763 | **0.767** | **0.767** | **0.763** | **0.771** | **0.765** | **0.767** | **0.765** | 0.763 | 0.761 |
| Random Filter | 0.739 | 0.731 | 0.725 | 0.729 | 0.728 | 0.728 | 0.734 | 0.728 | 0.733 | 0.730 | 0.733 | 0.725 |
| Activation Defense | 0.716 | 0.722 | 0.708 | 0.697 | 0.727 | 0.711 | 0.731 | 0.721 | 0.708 | 0.718 | 0.704 | 0.516 |
| Spectral Signatures | 0.682 | 0.644 | 0.680 | 0.692 | 0.674 | 0.686 | 0.703 | 0.690 | 0.492 | 0.672 | 0.687 | 0.689 |
| ISPL | 0.674 | 0.669 | 0.666 | 0.664 | 0.666 | 0.675 | 0.677 | 0.683 | 0.667 | 0.677 | 0.657 | 0.674 |
| Data Augmentation | 0.748 | 0.742 | 0.745 | 0.741 | 0.747 | 0.728 | 0.756 | 0.745 | 0.749 | 0.751 | 0.729 | 0.740 |
| Adversarial Training | 0.303 | 0.326 | 0.334 | 0.315 | 0.326 | 0.283 | 0.310 | 0.350 | 0.347 | 0.339 | 0.334 | 0.311 |
| D.A. + A.T. | 0.402 | 0.408 | 0.396 | 0.398 | 0.411 | 0.406 | 0.380 | 0.342 | 0.414 | 0.373 | 0.391 | 0.391 |

| Model | Witches' Brew | | | |
|---|---|---|---|---|
| | 0% | 10% | 20% | 30% |
| Undefended | 0.637 | 0.634 | 0.632 | 0.627 |
| Perfect Filter | **0.771** | **0.738** | **0.731** | **0.716** |
| Random Filter | 0.734 | 0.710 | 0.703 | 0.701 |
| Activation Defense | 0.731 | 0.548 | 0.641 | 0.637 |
| Spectral Signatures | 0.703 | 0.668 | 0.658 | 0.656 |
| ISPL | 0.710 | 0.711 | 0.694 | 0.690 |
| Data Augmentation | 0.643 | 0.629 | 0.621 | 0.613 |
| Adversarial Training | 0.280 | 0.270 | 0.260 | 0.254 |
| D.A. + A.T. | 0.315 | 0.307 | 0.305 | 0.315 |

Table 6: The attack success rate for the baselines and defenses evaluated on the CIFAR-10 dataset against the DLBD attack with the copyright and watermark triggers (top) and Witches' Brew attack (bottom) across varying poison percentages. The best performing models are in bold.

| Model | DLBD Copyright | | | | | DLBD Watermark | | | | |
|---|---|---|---|---|---|---|---|---|---|---|
| | 1% | 5% | 10% | 20% | 30% | 1% | 5% | 10% | 20% | 30% |
| Undefended | 0.228 | 0.624 | 0.777 | 0.866 | 0.879 | 0.340 | 0.764 | 0.869 | 0.939 | 0.947 |
| Perfect Filter | 0.051 | 0.094 | 0.081 | 0.067 | **0.081** | 0.033 | **0.025** | **0.026** | **0.030** | **0.027** |
| Random Filter | 0.238 | 0.623 | 0.713 | 0.847 | 0.881 | 0.258 | 0.703 | 0.812 | 0.907 | 0.944 |
| Activation Defense | 0.249 | 0.475 | 0.694 | 0.784 | 0.897 | 0.212 | 0.978 | 0.603 | 0.789 | 0.636 |
| Spectral Signatures | 0.225 | 0.451 | 0.606 | 0.687 | 0.828 | 0.121 | 0.103 | 0.581 | 0.831 | 0.601 |
| ISPL | 0.041 | 0.060 | 0.091 | 0.215 | 0.646 | 0.027 | 0.034 | 0.082 | 0.284 | 0.602 |
| Data Augmentation | 0.051 | 0.066 | 0.087 | 0.272 | 0.456 | 0.285 | 0.650 | 0.815 | 0.900 | 0.943 |
| Adversarial Training | **0.014** | **0.011** | **0.015** | **0.040** | 0.133 | 0.043 | 0.372 | 0.668 | 0.114 | 0.424 |
| D.A. + A.T. | 0.026 | 0.028 | 0.023 | 0.069 | 0.093 | **0.010** | 0.470 | 0.119 | 0.742 | 0.764 |

| Model | Witches' Brew | | |
|---|---|---|---|
| | 10% | 20% | 30% |
| Undefended | 0.167 | 0.300 | 0.433 |
| Perfect Filter | 0.033 | 0.033 | 0.067 |
| Random Filter | 0.633 | 0.600 | 0.667 |
| Activation Defense | 0.400 | 0.667 | 0.733 |
| Spectral Signatures | 0.700 | 0.733 | 0.600 |
| ISPL | 0.533 | 0.467 | 0.667 |
| Data Augmentation | 0.467 | 0.567 | 0.600 |
| Adversarial Training | 0.067 | 0.100 | 0.000 |
| D.A. + A.T. | **0.033** | **0.033** | **0.000** |

