# OpenReview forum: "Benchmarking the Effect of Poisoning Defenses on the Security and Bias of the Final Model"
_NeurIPS.cc/2022/Workshop/TSRML — TSRML2022_

### Official Review · Reviewer_MRaZ · 2022-10-18
**A relevant study with useful results**

**Overall Rating:** 6

**Summary:**

The paper conducts a comprehensive study of defenses against adversarial poisoning and tries to provide an additional perspective on the effect of these defenses. The authors evaluate the bias introduced into the treatment of different sub-populations in the dataset using statistical parity difference (SPD). The split into sub-populations is performed via BEAN regularization.  They argue why SPD is an important performance metric in the poisoning domain.

**Strengths:**

The authors cover a sufficiently big set of poisoning modes (DLBD/CLBD), triggers (Peace Sign/Bullet hole), poisoning percentages, defenses and datasets (CIFAR10/GTSRB) to demonstrate informative benchmarking results. They formulate and motivate a novel metric based on statistical parity difference.

Moreover, the authors introduce several straightforward "defenses" to give an additional perspective on their evaluation. They rigorously describe the setting and the considered methods. Overall the goal of providing a meaningful benchmark/survey for the field of defenses against poisoning seems to be achieved.

**Weaknesses:**

Not necessarily a weakness, but it would be interesting to see the ablation of the SPD metric with other sub-population generation than BEAN since this step seems to be crucial for this contribution.

How do you explain the reason why some defenses are skewed towards out-of-distribution data (Population 2)?

**Overall Recommendation:**

The authors provide a comprehensive benchmark for the poisoning defenses and introduce a new metric to evaluate their performance. Given a convincing discussion and a wide range experimental results, I would recommend accepting this paper.

**Review Confidence:**

4: The reviewer is confident but not absolutely certain that the evaluation is correct

---

### Official Review · Reviewer_vZUh · 2022-10-19

**Overall Rating:** 8

**Summary:**

This paper investigates the effects of poisoning defenses on fairness, using statistical parity difference (SPD) as a quantitative metric. Empirical evaluation is done on different defense methods, with comprehensive ablation studies.

**Strengths:**

It is meaningful to check the side effects of existing or future defense methods since almost all of the defense strategies implicitly introduce inductive biases. This paper investigate the aspect of fairness by using SPD, and provide comprehensive empirical studies on several defense methods.

**Weaknesses:**

Technically this paper is well-qualified as a workshop paper. To be extended to a full paper, the authors may involve more diverse empirical evaluation, and polish the figures (e.g., it is difficult to match different methods with different lines in Figure 1).

**Overall Recommendation:**

A good paper studies the side effect (from the fairness aspect) of poisoning defenses.

**Review Confidence:**

4: The reviewer is confident but not absolutely certain that the evaluation is correct

---

### Official Review · Reviewer_ZyrS · 2022-10-21
**I have concerns but the paper remains valuable given its comprehensive experiments**

**Overall Rating:** 7

**Summary:**

For each class in a test set, they first use an explanatory BEAN regularization[6] to divide it into a 'well-represented' group and a 'out-of-distribution' group.

They propose to use the difference of accuracy on these two groups (for different classes) as a fairness metric to assess whether existing defenses to data poisoning results in less fair models.

They evaluate a wide range of defenses using the proposed metric.


**Strengths:**

1. The topics, data poisoning & fairness, are both important ones and thus this should be interesting to our community.
2. Very comprehensive evaluations where a diverse set of defenses are included.
3. It is overall well-written.

**Weaknesses:**

1. I have concerns regarding identifying sub-populations by using a explanatory model to divide the test samples (in each class) into two groups, the 'well-represented' one and the 'out-of-distribution' one.

Naturally there are harder test samples and there are easier test samples (with respect to a given method). How is the gap between accuracy on hard&easy samples a fairness issue? Is it possible that the metrics we computed over these groups (identified by the explanatory model) reflect more on how similar the evaluated model is to the explanatory model?

I believe the metric will be more interpretable and more related to fairness if the groups are identified using natural, pre-defined concepts. An example (in the case of traffic sign recognition) will be the brightness mentioned in Appendix A.1, however I do not see that this example is strongly related with the proposed approach for identifying sub-populations.

2. This is more of a discussion: With the current formulations of the two groups, is it possible that the gap will be inevitably enlarged when we get stronger defenses? This can be another reason to use natural, pre-defined concepts to identify sub-populations.

There is this recent paper, Lethal Dose Conjecture on Data Poisoning (https://arxiv.org/abs/2208.03309), conjecturing that the sample complexity to predict a (test) sample correctly is inversely proportional to the fraction of poisoned samples one can possibly tolerated, i.e. a harder sample will be naturally more vulnerable to poisoning attacks. If this conjecture is true (and they did show that it is true at least in some cases), then it seems reasonable to me when a more robust defense performs worse on some harder samples compared to less robust ones, and that can be associated with the SPD metric used in this paper.

**Overall Recommendation:**

I have some concerns regarding the proposed method for identifying sub-populations from the test set. Even so, I believe the paper is still valuable and can interest many from our community, given its comprehensive experimental results.

**Review Confidence:**

4: The reviewer is confident but not absolutely certain that the evaluation is correct

---

### Decision · Program_Chairs · 2022-10-23

**Decision:**

Accept

**Comment:**

Great working connecting poisoning, robustness and fairness.